Ankle and midtarsal joint quasi-stiffness during walking with added mass

Kern Andrew M. 1 andrewkern@unomaha.edu
Papachatzis Nikolaos 1
http://orcid.org/0000-0003-4294-1557 Patterson Jeffrey M. 1
http://orcid.org/0000-0003-3815-6089 Bruening Dustin A. 2
Takahashi Kota Z. 1 ktakahashi@unomaha.edu
1 Department of Biomechanics, University of Nebraska at Omaha , Omaha, NE , USA
2 Exercise Sciences Department, Brigham Young University , Provo, UT , USA
Abdala Virginia
Electronic publication date: 2019 Sep 19
Publication date: 2019
Volume: 7
Electronic Location ID: e7487
Received 2018 Dec 3; Accepted 2019 Jul 16
Copyright: © 2019 Kern et al.
Copyright year: 2019
Copyright holder: Kern et al.
License: This is an open access article distributed under the terms of the Creative Commons Attribution License, which permits unrestricted use, distribution, reproduction and adaptation in any medium and for any purpose provided that it is properly attributed. For attribution, the original author(s), title, publication source (PeerJ) and either DOI or URL of the article must be cited.
License URL: https://creativecommons.org/licenses/by/4.0/

Keywords: Arch, Locomotion, Foot, Biomechanics, Windlass mechanism, Midtarsal joint, Ankle joint

Funding: University of Nebraska Collaboration Initiative, The Center for Research in Human Movement Variability of University of Nebraska at Omaha, NIH P20GM109090 National Institute On Aging of the National Institutes of Health F31AG057136 This work was supported by the University of Nebraska Collaboration Initiative, the Center for Research in Human Movement Variability of University of Nebraska at Omaha, NIH (P20GM109090), and the National Institute On Aging of the National Institutes of Health (F31AG057136). The funders had no role in study design, data collection and analysis, decision to publish, or preparation of the manuscript.

==============================
Examination of how the ankle and midtarsal joints modulate stiffness in response to increased force demand will aid understanding of overall limb function and inform the development of bio-inspired assistive and robotic devices. The purpose of this study is to identify how ankle and midtarsal joint quasi-stiffness are affected by added body mass during over-ground walking. Healthy participants walked barefoot over-ground at 1.25 m/s wearing a weighted vest with 0%, 15% and 30% additional body mass. The effect of added mass was investigated on ankle and midtarsal joint range of motion (ROM), peak moment and quasi-stiffness. Joint quasi-stiffness was broken into two phases, dorsiflexion (DF) and plantarflexion (PF), representing approximately linear regions of their moment-angle curve. Added mass significantly increased ankle joint quasi-stiffness in DF (p < 0.001) and PF (p < 0.001), as well as midtarsal joint quasi-stiffness in DF (p < 0.006) and PF (p < 0.001). Notably, the midtarsal joint quasi-stiffness during DF was ~2.5 times higher than that of the ankle joint. The increase in midtarsal quasi-stiffness when walking with added mass could not be explained by the windlass mechanism, as the ROM of the metatarsophalangeal joints was not correlated with midtarsal joint quasi-stiffness (r = −0.142, p = 0.540). The likely source for the quasi-stiffness modulation may be from active foot muscles, however, future research is needed to confirm which anatomical structures (passive or active) contribute to the overall joint quasi-stiffness across locomotor tasks.

Introduction

Humans dynamically alter the stiffness of their limbs in response to environmental and functional demands. Leg stiffness is modulated during locomotion to control multiple important biomechanical factors including: center of mass excursion when stepping between surfaces of differing compliance (Ferris, Liang & Farley, 1999), resistance to perturbation when walking on uneven terrain (Voloshina & Ferris, 2015) and changing stride frequency (Farley & González, 1996). Alternatively, understanding the stiffness behavior of individual joints (i.e., the hip, knee, ankle and foot) may enable better understanding of overall locomotor function (Günther & Blickhan, 2002). Ankle stiffness has been implicated as a primary determinant of overall leg stiffness during hopping (Farley & Morgenroth, 1999), and is an important area of investigation for overall walking performance. Understanding foot and ankle behavior during walking may be important for identifying gait pathology, human performance or improving prosthetic and assistive device design.

The ankle/foot complex performs multiple functional roles during gait, including support of body weight (Ogamba et al., 2016), acting as a lever for propulsion (Bojsen-Møller, 1979) and adaptation to environmental alterations (Voloshina & Ferris, 2015). Under normal walking, the ankle joint is the largest contributor of positive work (~40–50%) (Farris & Sawicki, 2012), however, structures distal to the ankle (i.e., foot) display negative net work (Takahashi, Worster & Bruening, 2017). Although the foot itself has been reported to lose energy (Takahashi, Worster & Bruening, 2017), its function is an important factor in gait performance. During push-off, the foot behaves as a lever to provide a mechanical advantage with the ground, where a stiffer foot allows the center of pressure (COP) to be placed further anteriorly to the ankle (Takahashi et al., 2016). Therefore, foot stiffness may alter the lever arm between the ankle and COP during stance, and allow modulation of the ankle’s mechanical advantage (Takahashi et al., 2016).

Quasi-stiffness of the ankle (sometimes called dynamic stiffness) is defined as the slope of the joints’ moment-angle relationship (Sanchis-Sales et al., 2016; Shamaei, Sawicki & Dollar, 2013; Rouse et al., 2013). This is an experimentally derived parameter, which describes the joints’ resistance to motion for a given change in moment throughout stance. Although not fully explored, a joint’s quasi-stiffness is likely influenced by active components, in which muscular activation surrounding the joint can dynamically control resistance to motion. In performing this analysis, near-linear regions of the joint moment-angle curve are identified and the linear slope of these regions are utilized to characterize overall joint quasi-stiffness (Sanchis-Sales et al., 2016).

Since the ankle accounts for a large proportion of total leg stiffness behavior during hopping (Farley & Morgenroth, 1999), and the ankle is a major contributor of positive work during gait (Farris & Sawicki, 2012), it is critical to understand its quasi-stiffness across a multitude of locomotor tasks. Ankle quasi-stiffness is highly variable across subjects and walking conditions, and increases with walking speed (Shamaei, Sawicki & Dollar, 2013; Collins et al., 2018). Additionally, the ankle’s moment-angle relationship becomes increasingly non-linear as walking speed increases (Hansen et al., 2004), and when varied locomotor tasks are performed (e.g., stair descent, stair ascent, walking and running) (Argunsah Bayram & Bayram, 2018). Although the quasi-stiffness of the ankle has been explored across a wide array of activities, the quasi-stiffness behavior of the entire foot and ankle complex is not well understood.

There are a limited number of studies which have investigated the quasi-stiffness of joints within the foot during locomotion, despite its potentially large influence on ankle behavior. During human walking, Sanchis-Sales et al. (2016) quantified quasi-stiffness of the ankle, midtarsal and the metatarsophalangeal (MTP) joints of healthy participants. Quasi-stiffness in the ankle and MTP joints was found to be lower than the midtarsal joint. In addition, the quasi-stiffness of the ankle, midtarsal and MTP joints varied within subjects based on the phase of stance (Sanchis-Sales et al., 2016).

There are several factors which may modulate midtarsal quasi-stiffness including the windlass mechanism, variations in foot structure (standing arch height) and utilization of intrinsic foot muscles. The windlass mechanism operates when the plantar fascia wraps around the head of the metatarsal bones and is tensioned by extension of the MTP joint (Hicks, 1954; Holowka & Lieberman, 2018; Welte et al., 2018). This allows motion in the MTP joint to affect properties of the midtarsal joint. Conversely, standing arch height has been shown to affect leg quasi-stiffness (Powell, Queen & Williams, 2016) and ankle quasi-stiffness (Powell et al., 2014) where a higher arch is related to greater stiffness. Finally, there is a substantial body of evidence which shows that activation of intrinsic foot muscles play an important role in resisting deformation of the arch during loading (Kelly et al., 2014), influencing energy storage and return from the surrounding elastic structures (Kelly et al., 2019) and adapting to various locomotion demands (Riddick, Farris & Kelly, 2019). Despite preliminary understanding of how midtarsal stiffness may be modulated, there is limited investigation of how the ankle/foot complex quasi-stiffness changes across varied dynamic tasks (e.g., walking with altered force demand or terrain).

The purpose of this study was to quantify the effect of increased body mass on the quasi-stiffness of the ankle and midtarsal joints during walking. The challenge of increased body mass was selected to directly increase the forces experienced by the foot and ankle (Huang & Kuo, 2014), which may trigger alteration of joint function. When walking with added mass, previous work has shown that ankle joint kinematics remain relatively unchanged, while ankle moment and power increase substantially (Huang & Kuo, 2014). It was hypothesized that increased body mass would lead to increased quasi-stiffness in both the ankle and midtarsal joints, by primarily increasing joint moment while joint angle remains relatively unchanged. Additionally, a secondary analysis was performed in an attempt to understand mechanisms implicated in midtarsal quasi-stiffness modulation, namely, the windlass mechanism and standing arch height. Such findings may improve our understanding of the compliance/rigidity relationship that the ankle/foot complex uses to adapt to external environments. This understanding could lead to the development of novel prostheses or assistive devices that allow for adaptation to tasks of daily living.

Materials and Methods

A total of 21 participants (15 males and 6 females, ages 24.1 ± 2.9 years, body mass 83.2 ± 20.3 kg, height 173.6 ± 6.7 cm; mean ± standard deviation) were recruited under the oversight of the University of Nebraska Medical Center Institutional Review Board (# 757-16-EP). All subjects gave written consent to participate in this study. All participants were healthy (no history of cardiovascular, neurological or musculoskeletal problems), with no history of impaired mobility. Individuals who were pregnant, had lower extremity pain, injury or surgery within the last year were excluded from this study. Medical history and impairment were assessed through the use of a medical history questionnaire.

Subjects walked over-ground while wearing three different levels of added mass: 0%, 15% and 30% of body mass. Subjects walked barefoot at 1.25 m/s in a straight line for ~10 m. Mass was added to subjects using a vest, which was loaded with 2.5 kg metal weights equally distributed medial-laterally and anterior-posteriorly. Targeted walking velocity was ensured using an optical timing system (Dashr, Lincoln, NE, USA) where only trials which had an average velocity within 1.20 and 1.30 m/s were accepted. Trials with differing levels of body mass were conducted in a randomized order to minimize any potential bias caused by fatigue or acclimatization to the protocol.

Foot and ankle kinematics were recorded using an 8-camera motion capture setup (Raptor-4s; Motion Analysis Corporation, Santa Rosa, CA, USA) captured at 180 Hz. A set of 53 retro-reflective markers were positioned on the subjects’ lower extremities. The shank, hindfoot, forefoot and hallux segments were identified bi-laterally using a marker set defined previously published multi-segment foot model (Bruening, Cooney & Buczek, 2012a. 2012b). The hallux segment spans from the hallux marker to the first MTP joint. The forefoot segment spans from the first MTP joint to the midtarsal joint (identified between markers placed on the navicular and cuboid bones). The hindfoot segment spans from the midtarsal joint to the most posterior aspect of the calcaneus. The hindfoot is connected to the shank segment via the ankle joint proximally, where the ankle joint center is defined as the midpoint between markers on the lateral and medial malleoli. Kinetic data was measured using a series of five force plates (400 × 600 × 82.55 mm; AMTI Inc., Watertown, MA, USA) located midway along a 10-m path and captured at 1,080 Hz. During collection, only trials in which the subject’s right foot fell entirely within the perimeter of a single force plate were accepted.

Following collection, data was processed using Visual3D software (C-Motion Inc., Germantown, MD, USA) where data was smoothed using a second order dual-pass low-pass Butterworth filter at 6 Hz for kinematic data and 25 Hz for kinetic data.

Moments about the midtarsal and ankle joints were quantified by computing the cross product of the ground reaction force with a vector from the joint center to the COP of that limb. This method was chosen over inverse dynamics because as many as three foot segments were simultaneously contacting the force plate. In the midtarsal joint, joint moment was computed only when the COP was anterior to the joint center, and zero moment was assumed when COP was posterior to the midtarsal joint center (Bruening & Takahashi, 2018). MTP joint kinetics were not characterized in this study, as MTP joint moment has been shown to have large amount of error using this methodology (Bruening & Takahashi, 2018). Ankle and midtarsal joint angles were determined by the angle between the shank and hindfoot segments, and the hindfoot and forefoot segments, respectively. Joint angles were not normalized relative to their standing calibration trial.

Joint moment data (normalized to biological body mass) and angle data were processed into quasi-stiffness values using a custom script written in MATLAB (Mathworks, Natick, MA, USA). This script identifies two functional phases within stance, dorsiflexion (DF) and plantarflexion (PF). DF phase spans from the instant of peak PF to peak DF in the ankle joint. As midtarsal joint moment was assumed to be zero until the joint COP was anterior to the joint center, DF phase in the midtarsal joint began when a non-zero joint moment was present. PF phase spans from peak DF to toe off at both the ankle and midtarsal joints. Quasi-stiffness was quantified by the slope of the moment-angle curve (Fig. 1) of both the ankle and midtarsal joints (Shamaei, Sawicki & Dollar, 2013) within these predefined regions (Fig. 2).

Figure 1 Ankle and midtarsal joint quasi-stiffness across all levels of added mass.

Average moment-angle curves for all 21 subjects for ankle (A) and midtarsal (B) joints. Joint quasi-stiffness was quantified in each subject by performing a linear fit in two phases of stance (dorsiflexion and plantarflexion) for each joint. The slope of each linear fit is reported as the quasi-stiffness value.

Figure 2 Ankle, midtarsal and metatarsophalangeal joint dorsiflexion angles during dorsiflexion and plantarflexion.

Ankle (A), midtarsal (B) and metatarsophalangeal (MTP) joint (C) dorsiflexion angles. Dark gray overlay shows dorsiflexion phase for ankle (A) and midtarsal joint (B and C). Light gray overlay shows plantarflexion phase for ankle (A) and midtarsal joint (B and C).

Quasi-stiffness of the ankle and midtarsal joints were computed for each subject using the average moment-angle curve of all trials for each weight condition (0%, 15% and 30% added body mass). There were between four and eight valid trials remaining for each subject and each condition after eliminating trials with errors in marker tracking (large gaps in foot marker positions during stance), walking speed, or force plate contact. A one-way repeated measures ANOVA was utilized to examine the effects of added body mass to changes in quasi-stiffness, range of motion (ROM) and peak moment in the ankle and midtarsal joints using JASP software (JASP Team, https://jasp-stats.org/). All kinetic variables (quasi-stiffness and moment) were normalized to the body mass of each subject at the 0% added mass condition. All study results were examined for sphericity using a test of Mauchly’s W. If a significant deviation from sphericity was found, a Greenhouse–Geisser correction was performed. If a main effect was found, pairwise comparisons were performed using a Holm–Bonferroni post hoc test.

Secondary analysis

A secondary analysis was conducted to better understand the mechanical factors that influence midtarsal joint quasi-stiffness. The first analysis was designed to measure the influence of the windlass mechanism on overall quasi-stiffness. The second measured the influence of standing arch height on changes in midtarsal stiffness due to added mass.

Utilization of the windlass mechanism was measured by quantifying the ROM of the MTP joint either within the DF or PF phase of the midtarsal joint (Fig. 2). A larger utilization of the windlass mechanism corresponds to larger MTP ROM. MTP joint angle was defined as the angle between the hallux and forefoot segments (Bruening, Cooney & Buczek, 2012a). MTP ROM was computed within the bounds of midtarsal DF and the entirety stance for the 0% and 30% added mass condition. This was compared with overall midtarsal quasi-stiffness in two ways: (1) between subjects, as a linear correlation between normalized quasi-midtarsal stiffness and MTP ROM at 0% body mass, and (2) within subjects, as a linear correlation between change in midtarsal quasi-stiffness with increased body mass (30–0%), and change in MTP ROM.

Standing arch height was approximated using midtarsal joint angles obtained during static calibration trials at 0% added mass. Static standing arch height is an additional metric which may explain the variation in midtarsal quasi-stiffness seen between subjects in this study. A linear correlation was used to examine the relationship between standing arch height and midtarsal joint stiffness in the 0% added body mass during the midtarsal DF and PF phases.

Results

Ankle joint

The added mass condition displayed no significant effect (F(2,40) = 0.822, p = 0.447) on ankle ROM (0%: 19.8° ± 3.6°, 15%: 20.1° ± 3.7°, 30%: 19.74° ± 3.2°, mean ± std). However, peak PF moment (0%: 1.48 ± 0.15 Nm/kg, 15%: 1.61 ± 0.16 Nm/kg, 30%: 1.72 ± 0.18 Nm/kg, mean ± std) significantly increased with added mass (F(2,40) = 279.1, p < 0.001) (Fig. 3). Post hoc tests found a significant difference in peak ankle moment between all levels of added mass (p < 0.001). Similarly, it was found that added mass significantly increased joint quasi-stiffness, during both the DF (F(2,40) = 33.27, p < 0.001), (0%: 0.089 ± 0.02 Nm/kg°, 15%: 0.093 ± 0.02 Nm/kg°, 30%: 0.100 ± 0.02 Nm/kg°, mean ± std) and PF (F(2,40) = 21.48, p < 0.001), (0%: 0.091 ± 0.014 Nm/kg°, 15%: 0.099 ± 0.020 Nm/kg°, 30%: 0.107 ± 0.018 Nm/kg°, mean ± std) phases (Fig. 4). During the DF and PF phase respectively, post hoc tests revealed a significant difference (p ≤ 0.013 and p ≤ 0.018) between all weight conditions (Fig. 4).

Figure 3 Ankle moment, angle, range of motion and peak moment for added body mass conditions.

Ankle dorsiflexion angle (A), range of motion (B), plantarflexion moment (C), and peak moment (D). Added body mass significantly increased peak moment (p < 0.001), but not ankle range of motion (p = 0.447). Brackets indicate significance of pairwise tests (p < 0.05).

Figure 4 Ankle and midtarsal joint stiffness during dorsiflexion and plantarflexion.

Ankle (A) and midtarsal (B) quasi-stiffness across dorsiflexion (DF) and plantarflexion (PF) phases for all three levels of added mass (0%: blue, 15%: green, 30%: red). A significant effect of increased body mass on quasi-stiffness was found on the ankle in DF (p < 0.001) and PF (p < 0.001), and in the midtarsal joint in both DF (p = 0.006) and PF (p < 0.001). Brackets indicate significance of pairwise tests (p < 0.05).

Midtarsal joint

The added mass condition displayed no significant effect (F(2,40) = 1.335, p = 0.275) on midtarsal ROM (0%: 12.7° ± 3.2°, 15%: 12.9° ± 3.3°, 30%: 13.1° ± 3.2°, mean ± std). There was, however, a significant increase (F(1.370, 27.398) = 256.3, p < 0.001) on peak PF moment (0%: 1.098 ± 0.136 Nm/kg, 15%: 1.197 ± 0.146 Nm/kg, 30%: 1.278 ± 0.161 Nm/kg, mean ± std) (Fig. 5). Post hoc tests found a significant difference in peak midtarsal moment between all levels of added mass (p < 0.001). Added mass had a significant effect on joint quasi-stiffness in the DF phase (F(2,40) = 5.742, p = 0.006), (0%: 0.23 ± 0.1 Nm/kg°, 15%: 0.25 ± 0.11 Nm/kg°, 30%: 0.26 ± 0.12 Nm/kg°, mean ± std) as well as PF phase (F(2,40) = 15.24, p < 0.001), (0%: 0.093 ± 0.024 Nm/kg°, 15%: 0.101 ± 0.026 Nm/kg°, 30%: 0.104 ± 0.026 Nm/kg°, mean ± std) in midtarsal quasi-stiffness (Fig. 4). During the DF phase, post hoc analysis revealed a significant difference only between the 0–15% (p = 0.040) and 0–30% (p = 0.006) conditions. During the PF phase, post hoc analysis revealed a significant difference only between the 0–15% (p = 0.006) and 0–30% (p < 0.001) conditions (Fig. 4).

Figure 5 Midtarsal moment, angle, range of motion and peak moment for added body mass conditions.

Midtarsal dorsiflexion angle (A), range of motion (B), plantarflexion moment (C), and peak moment (D). Added body mass significantly increased peak moment (p < 0.001), but not range of motion (p = 0.275). Brackets indicate significance of pairwise tests (p < 0.05).

Secondary analysis

No correlation (r = 0.181, p = 0.431) was found between MTP joint ROM and midtarsal joint stiffness at 0% added mass during midtarsal joint DF. Similarly, no correlation was found between change (30–0% added mass) in MTP ROM, and change in midtarsal DF (r = −0.142, p = 0.54) stiffness (Fig. 6). No correlation (r = −0.214, p = 0.352) was found between MTP joint ROM and midtarsal joint stiffness at 0% added mass during midtarsal joint PF. No correlation was found between change (30–0% added mass) in MTP ROM, and change in midtarsal PF (r = −0.352, p = 0.118) stiffness (Fig. 7). No significant correlation was found between standing arch height and change in midtarsal stiffness, in either the DF (r = −0.158, p = 0.494) or PF (r = −0.146, p = 0.527) phases (Fig. 8).

Figure 6 Secondary analysis of metatarsophalangeal joint ROM and midtarsal joint stiffness during dorsiflexion.

Secondary analysis examining the relationship between midtarsal quasi-stiffness and metatarsophalangeal joint (MTP) range of motion (ROM) during midtarsal joint dorsiflexion phase. MTP ROM is defined as the difference between the minimum and maximum joint angle within the dorsiflexion phase. No significant relationship was found between MTP ROM and MT joint stiffness (A) or between change in MTP ROM and change in MT joint stiffness (B), suggesting that factors other than MTP dorsiflexion (and the windlass mechanism) were important in modulating MT joint stiffness.

Figure 7 Secondary analysis of metatarsophalangeal joint ROM and midtarsal joint stiffness during plantarflexion.

Secondary analysis examining the relationship between midtarsal quasi-stiffness and metatarsophalangeal joint (MTP) range of motion (ROM) during midtarsal joint plantarflexion phase. MTP ROM is defined as the difference between the minimum and maximum joint angle across stance. No significant relationship was found between MTP ROM and MT joint stiffness (A) or between change in MTP ROM and change in MT joint stiffness (B).

Figure 8 Secondary analysis of midtarsal joint standing angle and midtarsal quasi-stiffness in dorsiflexion and plantarflexion.

Secondary analysis exploring the relationship between midtarsal quasi-stiffness and midtarsal standing angle (arch height) during dorsiflexion phase (A) and plantarflexion phase (B). There was no significant relationship between standing midtarsal angle and midtarsal quasi-stiffness.

Discussion

The overall goal of this study was to explore the effect of added body mass on ankle and midtarsal joint quasi-stiffness during walking. Our hypothesis that ankle quasi-stiffness increases with added mass was supported in both DF and PF phases. Our hypothesis that midtarsal quasi-stiffness increases with added mass was also supported in the DF and PF phases. These findings support the mounting evidence that biological limbs can modulate stiffness in response to varying locomotor demands (Ferris, Liang & Farley, 1999; Farley & Morgenroth, 1999; Argunsah Bayram & Bayram, 2018).

The moment produced at the ankle and midtarsal joints increased with added mass, but their ROM did not increase correspondingly, which is consistent with the overall trend of a stiffer joint and in agreement with our hypothesis. Overall, there appeared to be little change in either the foot or ankle kinematics with added mass. This is supported by other work which has found little change in ankle kinematics when walking with added mass distributed about the center of mass (Huang & Kuo, 2014) and loaded anteriorly (Ogamba et al., 2016). These results suggest that ankle quasi-stiffness modulation is not tightly coupled to absolute joint angle, a trait that may be beneficial for walking on varied surfaces. This also implies that ankle quasi-stiffness modulation requires at least some degree of muscular activation rather than purely passive mechanical properties as some active change is likely required to produce increased force from similar kinematics.

Midtarsal quasi-stiffness during the DF (mean quasi-stiffness of 0.23 ± 0.10 Nm/kg° for 0% added body mass) phase was substantially higher than that seen in the ankle or in the midtarsal joint during PF (mean quasi-stiffness of 0.093 ± 0.024 Nm/kg° for 0% added body mass). Although a precise mechanistic understanding for elevated midtarsal quasi-stiffness remains unclear, the values obtained in this study are in agreement with those conducted elsewhere. Sanchis-Sales et al. (2016) found midtarsal quasi-stiffness values in early stance to be 0.18 ± 0.28 Nm/kg° (converted from original units) which was higher and more variable than the ankle or MTP joint in their study when subjects were walking at their preferred walking speed. In Sanchis-Sales et al. (2016) study, ground reaction force was attributed to each foot segment (hindfoot, forefoot and hallux) through the use of a pressure sensitive mat, which may contribute to the minor differences seen in our data. In runners with rearfoot strike patterns, Bruening et al. (2018) found midtarsal joint quasi-stiffness values of 0.41 ± 0.11 Nm/kg°, which are understandably higher than the values in this study due to the more demanding running task. In order to explore mechanisms which might be associated with midtarsal quasi-stiffness, a secondary analysis was conducted.

Secondary analysis

Midtarsal joint quasi-stiffness during DF had high subject-to-subject and within-subject variability when compared to the ankle joint or to the midtarsal joint during PF. It is possible that the methodology used to compute midtarsal moment (assuming zero joint moment until the COP is anterior to the joint center) underestimated joint moments early in stance and biased results toward higher stiffness values. It seems unlikely however that this result caused a large effect on the overall results of the study, as a recent analysis has shown that only low magnitude joint moments are missed with this assumption (Bruening & Takahashi, 2018). There are several plausible mechanistic explanations for the high variability, which warrant further discussion including: standing arch height and utilization of the windlass mechanism.

Our secondary analysis examined the influence of standing midtarsal joint angle (used as a surrogate measure for standing arch height) on variability of the midtarsal quasi-stiffness. No significant correlation was found between standing midtarsal angle and midtarsal quasi-stiffness in DF or PF. This indicates that standing arch height did not play a major role in the midtarsal quasi-stiffness variability seen in this study. While this disagrees with previous work which has shown a relationship between arch height, leg stiffness (Powell, Queen & Williams, 2016) and dynamic ankle stiffness (Powell et al., 2014), it should be noted that several other studies have found only a weak (Zifchock et al., 2006), or no relationship (Holowka, Wallace & Lieberman, 2018) between standing arch height and foot stiffness. The use of standing midtarsal angle as a surrogate for standing arch height is un-validated using this marker set, and may be subject to significant errors from marker placement and foot length, perhaps confounding these findings. Therefore, future studies should explore the contribution of arch height on the variability of the midtarsal quasi-stiffness.

Utilization of the windlass mechanism was evaluated as an explanatory factor for the variability seen in midtarsal quasi-stiffness during the DF. We found no correlation between MTP joint ROM and midtarsal joint quasi-stiffness at 0% added body mass, and no correlation between change in MTP joint ROM and change in midtarsal joint quasi-stiffness (30–0% added mass). These findings suggest that utilization of the windlass mechanism did not account for between-subject variability, and did not explain increased midtarsal joint quasi-stiffness when walking with added mass. Our findings are in agreement with a recent study involving an isolated arch compression test, which showed that MTP joint DF did not increase overall arch stiffness (Welte et al., 2018).

A potential explanation for the increased midtarsal joint quasi-stiffness when walking with added mass may be activation of the foot muscles. Although electromyography was not collected in this study, recent studies have provided evidence that midtarsal mechanical behavior is influenced by activation of intrinsic foot muscles (abductor hallucis, flexor digitorum brevis or quadratus plantae) (Kelly et al., 2014) in which these muscles alter activations based on the locomotion task (Kelly, Lichtwark & Cresswell, 2015) or demand (Riddick, Farris & Kelly, 2019). However, another recent study found that an induced nerve block in the intrinsic foot muscles did not change the midtarsal quasi-stiffness during walking and running (Farris et al., 2019). Whether intrinsic foot muscles can contribute to midtarsal joint quasi-stiffness remains inconclusive and further research is needed to reconcile which anatomical structures are a major determinant of the midtarsal joint behavior.

Limitations

The range of added body mass that was tested on subjects may have been insufficient to trigger the full spectrum of possible changes seen in the foot and ankle. This study was conducted as a part of a larger study investigating foot mechanics and foot temperature. Therefore, additional trials were conducted either before or after collection of this data based on randomization of that specific subject. It is possible that fatigue or adaptation to the weight vest from that previous walking section may have confounded our results on a subset of subjects. However, given that prior trials were only three 10 min treadmill walking sessions, each separated by 30 min of rest, it seems unlikely that this would be a significant influence in a healthy young population.

Conclusions

This study sought to identify how ankle and midtarsal quasi-stiffness is influenced when healthy individuals walk with added mass. In general, it was found that ankle and midtarsal joint quasi-stiffness increases with added mass. Of particular importance was midtarsal joint DF, which has much higher quasi-stiffness than the ankle joint or the midtarsal joint during PF, and displayed large between- and within-subject variability. No significant relationship was found between utilization of the windlass mechanism (MTP joint ROM) and midtarsal joint quasi-stiffness, suggesting that factors outside of those measured (potentially activation of intrinsic foot muscles) in this study are important to overall joint stiffness. These findings are critical for improving our understanding of how the foot and ankle behaves during walking, running, load carriage and other physiologically realistic conditions, and for uncovering unique mechanical behavior of the foot and ankle. In addition, these data are useful in the design of biologically inspired prosthetic devices, which can replicate function of native joints across a wide variety of tasks of daily living.

Supplemental Information

Supplemental Information 1 Ankle and midtarsal range of motion, peak moment and stiffness.

Ankle and midtarsal range of motion, peak moment and stiffness and time series of normalized moment and angle.

Click here for additional data file.

Additional Information and Declarations

Competing Interests

Author Contributions

Human Ethics

Data Availability

The authors declare that they have no competing interests.

Andrew M. Kern analyzed the data, prepared figures and/or tables, authored or reviewed drafts of the paper, approved the final draft.

Nikolaos Papachatzis performed the experiments, analyzed the data, authored or reviewed drafts of the paper, approved the final draft.

Jeffrey M. Patterson performed the experiments, analyzed the data, authored or reviewed drafts of the paper, approved the final draft.

Dustin A. Bruening conceived and designed the experiments, analyzed the data, authored or reviewed drafts of the paper, approved the final draft.

Kota Z. Takahashi conceived and designed the experiments, performed the experiments, analyzed the data, authored or reviewed drafts of the paper, approved the final draft.

The following information was supplied relating to ethical approvals (i.e., approving body and any reference numbers):

The University of Nebraska Medical Center Institutional Review Board granted ethical approval to conduct this study (IRB # 757-16-EP).

The following information was supplied regarding data availability:

Ankle and midtarsal range of motion, peak moment, and quasi-stiffness during dorsiflexion and plantarflexion data and normalized time-series of ankle and midtarsal moment and ankle and midtarsal joint angle during stance are available as Supplemental Files.

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
