# Peer review of "Ankle and midtarsal joint quasi-stiffness during walking with added mass"

_PeerJ, doi:10.7717/peerj.7487_

## Round 0.1 · original submission · Major Revisions

First I would like to apologize for the long time it has taken to get back to you. I have now received two reviews of your paper. According to our first reviewer, you need to re-frame the questions that you are testing to make your results easier to interpret; you should reorganize the study, as some of your most important analyses are not mentioned until the discussion; you need to provide more detail in your methods, and should provide more of your raw data. Our second reviewer wonders wheter has your approach been applied previously for similar measures; also suggest that the discussion could be restructured; you should clarify if the variability reported in midfoot dorsiflexion quasi stiffness refers to stiffness between individuals, or in the response to increased load carriage within individuals; an alternative solution is suggested to determine the role of the Windlass in stiffening the mid-foot.

I would like to see all the reviewers’ suggestions taken into full consideration to improve your work.

Reviewer 1 ·

Basic reporting

This paper is clearly written with just a few minor grammatical errors. The paper provides a relatively thorough overview of research on human leg stiffness and joint quasi-stiffness during walking and running, although there is too much emphasis on leg stiffness during running that is not clearly relevant to the present study (see below). It also discusses results in the context of relevant research, although a more nuanced discussion of foot biomechanics models is warranted (see below). For the most part the article is structured appropriately, although the authors do not mention a significant component of their analysis until the discussion (see below), and thus do not present appropriate hypotheses or methods for this analysis. Outside of this, the results presented are appropriate. However, the authors should share more of their raw data, in particular data from individual subjects for all steps analyzed, including basic kinematic and kinetic waveform data used to calculate study variables.

Experimental design

This paper presents original research within the scope of PeerJ. The stated research question is to test the effects of added mass on quasi-stiffness of the ankle and midfoot joints, although the broader purpose of this goal is too vague. The reasoning behind the hypothesis of this study, that quasi-stiffness will increase with added mass, is also not explained. The study tests the research question thoroughly, although a few key details are missing from the methods (see below).

Validity of the findings

The results of the study appear robust, although the authors never state whether their data are actually normally distributed, which would determine the appropriateness of their use of repeated measures anovas. In case the data are not normally distributed for some tests, the authors could always used Wilcoxon Signed Rank tests. Conclusions are appropriate given data, although I have some issues with how the authors interpret their results with regard to the windlass mechanism and the midtarsal locking mechanism (see below).

Additional comments

In framing their research question, the authors discuss the literature on how leg stiffness is modulated during running in response to features of the terrain such as surface compliance and irregularity. However, these studies aren’t really relevant to the present research, as running is a mass spring gait in which using a specific leg stiffness is important to things like maintaining a constant CoM displacement and foot contact time. It is not clear how these things relate to walking, where leg stiffness is likely not constant throughout a step, and the importance of modulating leg stiffness is not as straightforward. On a related note, the authors point out that the ankle has been shown to be the most important contributor to overall leg stiffness, although to my knowledge this has only been demonstrated for mass-spring ‘gaits,’ like hopping and running. I recommend that the authors avoid discussing their research question in the context of overall leg stiffness during walking, and instead focus on how understanding the factors that dictate ankle and midfoot quasi-stiffness will improve our understanding of walking gait mechanics.

The authors do not provide a thorough enough description of what quasi-stiffness is, or why it’s important. They should provide a simple definition of quasi-stiffness, including how it’s calculated, and how it differs from other measurements of stiffness. They should also explain why quasi-stiffness is interesting and important for leg function, as well as what the broader implications of understanding quasi-stiffness is for things like prosthetics design and musculoskeletal health. More specifically, what do they see as the importance of quantifying quasi-stiffness with added mass?

The authors also need to provide some rationale for their hypotheses. Why should quasi-stiffness go up with added mass? The kinematic and kinetic data presented in Huang and Kuo (2014 JEB) could be used as a basis for predictions for the ankle. For the foot, the authors should draw on our current understanding of foot biomechanics. For this, they could use previous studies of foot kinetics or EMG (e.g. Sanchis-Sales et al. 2016 JAPMA; Kelly et al 2015 J R Interface), but should makes sure that their predictions are grounded in a biomechanical model of how the foot should function given added mass.

I believe that the authors should go a little further in interpreting their results as to the broader significance of this study. For instance, in finding that ankle and midtarsal stiffness goes up with added mass, what does this tell us about gait mechanics that we didn’t already know? Alternatively, what are the implications for prosthetic design, and/or musculoskeletal injury?

A major organizational issue in this study is that the authors do not mention testing the effects of the windlass mechanism and arch height on midtarsal joint quasi-stiffness until the discussion. This is an important part of this study, and the authors have included it in the abstract and conclusions. The authors should present predictions as to how the windlass mechanism and arch height will affect midtarsal quas-stiffness in the introduction, and describe all analyses and results for this part of the study in the appropriate sections. This is potentially a very interesting element of this study, and so needs to be set up and described properly. The authors are missing some key methodological detail concerning these analyses, including how they measured MTP joint angle, and statistics (are these data normally distributed?). They should also present a graph showing average MTP joint motion with relevant midtarsal dorsiflexion and plantarflexion quasi-stiffness phases highlighted.

In discussing the relevance of their findings to windlass mechanism, the authors should acknowledge that the contribution of the windlass mechanism to midtarsal dorsiflexion quasi-stiffness are likely to be small. Their analyses indicate that during this period, the MTP joints only undergo 0-10 degrees excursion, which may not have a big effect on tension in the plantar aponeurosis, which drives the windlass mechanism (Caravaggi et al 2010 JEB). The MTP dorsiflexion that occurs during plantarflexion quasi-stiffness may be more important. If so, these results suggest that the windlass mechanism does not have a major effect on midtarsal quasi-stiffness. However, the authors largely neglect intrinsic foot muscle activity as a possible major contributor to foot quasi-stiffness, as has been argued recently (e.g. Kelly et al., 2014, 2015, J R Interface). Although they can’t test this, they should discuss it as a possibility.

The authors suggest the midtarsal-locking mechanism may contribute to differences in midtarsal quasi-stiffness between participants, but do not test this idea. Considering they do not test it, and this model of human foot function has little empirical support (see Tweed et al., 2008 The Foot), I suggest that they do not discuss it.

Minor Issues:

lines 55-56 – Explain exactly what is meant by ‘structures distal to the foot.’

lines 57-59 – Explain more clearly how foot quasi-stiffness relates to moment arm length of the plantarflexors and ground reaction force, and what the implications are for gait (e.g. Takahashi et al 2016 Sci Rep).

Throughout, Sanchis-Sales et al (2016) is incorrectly cited as Sales et al. Please correct.

The authors should provide illustrations of the marker sets they used, how they defined kinetic foot segments, and where the joints between the foot segments are located. The description in the text is not clear enough. The authors also seem to be missing some detail about how pelvis and thigh segments are defined (only two markers/segment?), but this isn’t relevant to the present study.

lines 133-135 – The definition of the ankle joint doesn’t make sense. Why define it as motion between leg and forefoot segments, rather than leg and rearfoot segments? Or is this a typo?

lines 141-143 – What are the ankle and forefoot ‘rockers’?

The authors need to provide a much more thorough description of how they defined the ‘linear portions’ of the angle moment curves. Is there any way that they can ensure that these definitions were objective? This is another reason to include all of the raw data. Additionally, it would be useful to see the approximate timing of these phases on the moment and angle results figures, as in Sanchis-Sales et al (2016). At the very least, provide some indicate of the percentage of stance when these phases occur.

How many steps were analyzed per subject per condition?

line 173 – Change ‘ankle’ to ‘midtarsal’.

line 219 – Should this be quasi-stiffness in ‘dorsiflexion’ at midtarsal joint?

Standing midtarsal angle may not be a great proxy for arch height, as it is prone to error from marker placement, and it doesn’t take into account how much the arch deforms under weight-bearing relative to its height when unloaded. The authors should discuss this limitation of their study.

·

Basic reporting

The report is very well written, is concise, clear and unambiguous. There is adequate reference to the literature and the authors have included most of the relevant papers in this area. The authors have written a thorough introduction that justifies the need for the study.

The figures are well presented and well described.

The discussion could be restructured slightly. At present it is “joint quasi-stiffness” then “Kinematics and kinetics” then back to “midtarsal joint quasi-stiffness”. Perhaps aligning all the discussion around joint-quasi stiffness together might help with the flow of the discussion.

Experimental design

The research question is well defined and well justified. There is a clear gap in knowledge that this research addresses.

Methodology is generally appropriate and well designed. The study is adequately powered and the investigators should be commended on this.


Calculating quasi stiffness for the mid-foot and ankle during the loading and unloading phase seems unusual. Quasi stiffness would typically be measured during the loading (dorsiflexion) phase when negative work is being done at each respective joint. Given that stiffness is defined as the magnitude of deformation of a given applied force, calculating stiffness during unloading (recoil of the arch) is hard to define, as the mid-foot is not deforming, but rather returning to its original shape, as force declines. Has this approach been applied previously for similar measures?

In regard to the quasi stiffness measure of the mid-foot. I understand the difficulties associated with these calculations. Particularly with reference to assigning forces to various foot segments when multiple segments are in contact with the ground. This is particularly difficult in walking when the heel is on the ground for a considerable period of time. When I look at the angle-moment curves for the mid-foot I wonder if the large stiffness values for the mid-foot are an artifact of assuming zero moment until the COP passed the joint axis. There is obviously a moment generated about this joint prior to this point, as there is considerable midfoot dorsiflexion that occurs prior to the registration of any joint moment. I wonder if a simpler approach may be to fit a straight line between the midfoot angle at foot contact and the angle at peak midfoot moment? While this may seem less informative, it likely helps to avoid the pitfall of assuming the midfoot to be excessively stiff.

Validity of the findings

Refer to above comment pertaining to measures of quasi stiffness.

The authors report considerable variability in midfoot dorsiflexion quasi stiffness. Could they please clarify if this is variability in stiffness between individuals, or variability in the response to increased load carriage within individuals? If the former is the case, is it possible that this variability may be a consequence of the method used to quantify midfoot DF quasi-stiffness? See comment above about how assuming zero mid-foot joint moment until the COP progresses beyond the joint axis might lead to over-estimations in quasi-stiffness. Eg. those with faster COP progression under the foot will have reduced stiffness, as the joint moment is registered earlier in the stride cycle. Those with delayed COP progression under the foot have increased stiffness, as a considerable amount of midfoot dorsiflexion has already occurred before a joint moment is even calculated. This would be counter-intuitive, but might be an artifact of the calculation.

The additional investigation about the contribution of the windlass mechanism to midfoot stiffness is problematic. The Windlass mechanism is proposed to stiffen the arch in late stance, prior to propulsion. However, the data presented suggests no relationship between MTP joint range of motion and midfoot stiffness in the plantar flexion phase, when presumably the Windlass mechanism would be acting. In fact, the findings seem to contradict the effectiveness of the Windlass mechanism, as the midfoot is actually more complaint in the PF phase than the DF phase. This finding is also similar to the findings of Welte et al. (J. Royal Society interface, 2018). Although, this could also be related to the method of calculating midfoot quasi stiffness (see comments above). If the authors would like to determine the role of the Windlass in stiffening the mid-foot, an alternative solution might be to plot the MTP and midfoot joint angles and angular velocities. This may help to show that MTP joint rotation commences prior to midfoot plantarflexion, hence providing some evidence for the Windlass changing midfoot kinematics.

The authors describe large subject to subject variability in data. It would be fantastic to be able to present this data as supplementary or supporting data. For example, the authors could present angle-moment plots for the midfoot and ankle quasi stiffness for each individual, as supplementary data.

Additional comments

Specific comments
Line 142 – what are “the ankle and forefoot rockers”
Line 173 – should “peak ankle moment” be “peak midtarsal moment” ?
Line 186-187 – It is unclear what this sentence means. Please re-word to improve clarity
Line 193 – The moment is “produced” about a joint, rather than “experienced”
Line 237 – Why was the secondary analysis only performed on the 0% bodyweight condition?
Line 257 – Based on bone pin studies (see Nester et al. 2009) there really is no evidence to support the notion of mid-tarsal joint locking.

---

## Round 0.2 · Minor Revisions

I regret the delay in communicating my decision. Our reviewer is satisfied with the work that you have done to improve your manuscript. However, it needs a little more work; please, consider all these new suggestions before resending your study.

Reviewer 1 ·

Basic reporting

This article is generally well-written and organized, with a few minor grammatical errors throughout the text. I suggest the authors carefully re-read and edit the manuscript to remove these errors. The introduction and background set-up the goals and hypotheses of the study nicely, and are generally well-referenced. I have a few suggestions for additional literature that the authors may choose to reference below. The article is structured appropriately, and the figures are all excellent and easy to interpret. The authors now provide all of the source data that would be necessary to re-run analyses.

Experimental design

This study represents a novel contribution to our understanding of the factors that influence joint stiffness in the ankle and foot. The investigation is rigorous, and the methods are clearly described.

Validity of the findings

Results are valid and interesting, and interpretations are generally appropriate. I have one minor issue with the interpretation of the windlass mechanism results that I wish for the authors to consider (see below), but overall I think the conclusions of the study are warranted.

Additional comments

I believe the authors have done a fantastic job of revising this article, and feel that they have adequately addressed most of my previous issues. My one remaining issue is that I’m still skeptical that <10 degrees of MTP dorsiflexion could stiffen the foot appreciably via the windlass mechanism. I appreciate the argument extended by the authors regarding the findings of Caravaggi et al. 2010, but suspect that the 4% PA strain observed in that study was probably driven more by bending of the arch than MTP motion. Indeed McDonald et al. (2016 PlosOne 11: e0152602) found that during running, MTP motion accounted for less than 15% of the observed strain in the PA, with the remainder due to compression and the concomitant lengthening of the foot. While that study looked at running and not walking, I think it remains to be demonstrated that the relative contribution of the MTP joints to PA strain is substantially different in walking. I also have a hard time accepting that differences on the order of 2 degrees of MTP motion could play a substantial role in the observed differences in midtarsal stiffness between unweighted and weighted conditions. Nevertheless, the results of this study are difficult to explain otherwise, and so I think the authors have provided one possible interpretation. I think the authors should use these results to call for more research on the relative effects of small changes in MTP joint position on PA tension and midtarsal stiffness.

Concerning their findings on the absence of a relationship between arch height and midtarsal stiffness, it would be useful if the authors could provide some qualitative assessment of the range of arch heights they sampled. The arch height measurement method that they used provides values that are difficult to interpret, and the sources of error in this method (that they acknowledge) indicate that their findings can only be interpreted tentatively. Were there subjects in this study who could be classified as possessing ‘flexible flat foot’, or other extremes of the arch height spectrum? Their finding of a lack of relationship is surprising, since very high arches are anecdotally associated with rigid feet, and flat feet are anecdotally associated with highly compliant feet. Do these data allow the authors to contradict these common assumptions? This provides another opportunity to call for further research, especially research using arch height as a continuous variable.
I have a few more minor issues that I hope the authors will consider:
lines 52-60 - I think the explanation of quasi-stiffness provided here could use a little bit more clarification. In particular, it seems to me that part of what joint quasi-stiffness is telling us is how muscles are being used to resist motion at a joint. During gait, a joint does not have a single inherent quasi-stiffness value, but instead its stiffness will change based on how muscles are being used at a given time in stance. Thus, quasi-stiffness provides a measure of how muscle can be used to tune the stiffness of a joint for a given task (or for a given phase in gait.)
line 61 – This statement is too general. Farley and Morgenroth 1999 only found that the ankle was a major contributor to leg stiffness during hopping, but not any other tasks. Furthermore, they didn’t quantify quasi-stiffness, they quantified limb and ankle stiffness more generally. The authors should adjust this sentence accordingly.

lines 80-82 – Reference 22 (Zifchock et al) only shows a relationship between arch height and static arch stiffness, and this relationship is very weak. Furthermore, Holowka et al (2018 Sci Rep 8: 3679) found no correlation between static foot stiffness measurements and foot stiffness measurements during walking, so the connection between arch height and foot stiffness in the literature is tenuous at best. I think the authors should adjust these sentences accordingly.

There is an error in the authors names in reference 21.

---

## Round 0.3 · Minor Revisions

Thank you for your consideration of the reviewers' suggestions. I think that we are almost there, however, there are still a few points that need your attention. Please explain how exactly was ‘dorsiflexion phase’ determined for the midtarsal joint.

You should remove ‘quasi’ of the quasi-stiffness’ to described the ankle measurement used and thus to avoid giving readers the false impression that Farley and Morgenroth used a similar method of measuring stiffness as that used in this study. Please take these last suggestions of our reviewer in full consideration.

Reviewer 1 ·

Basic reporting

As stated in my previous review, I believe this article is well-written and organized. The grammatical errors have been corrected, and the article include appropriate references. The figures are excellent and appropriate. All source data is provided.

Experimental design

As previously stated, I think this study represents a novel contribution to our understanding of the factors that influence joint stiffness in the ankle and foot. The investigation is rigorous, and the methods are clearly described. I am pleased to see the authors have identified and amended an error in their prior analysis. I have one lingering issue about how they calculated midtarsal dorsiflexion stiffness, which I detail below.

Validity of the findings

I believe that the results are valid and interesting. Correcting the error in the prior analysis leads to results that make a lot more sense to me, and bolster some of the original conclusions of the study. While some conclusions have to be changed, these changes strengthen the findings of the paper in my view.

Additional comments

Fixing the analysis error in the prior submissions leads to results that make more intuitive sense to me, and so I commend the authors for identifying this issue and adjusting the paper accordingly. This makes the results and interpretation much more straightforward in my opinion. I just have one substantive issue remaining, which concerns the calculation of midtarsal joint dorsiflexion stiffness. In lines 148 to 149 the ‘dorsiflexion phase’ is defined as spanning the period from peak plantarflexion to peak dorsiflexion at each joint. However, for the midtarsal joint, Figure 2B shows this phase beginning at a somewhat arbitrary point, not at the peak joint plantarflexion. So, how exactly was ‘dorsiflexion phase’ determined for the midtarsal joint? Was the phase actually defined as beginning when the CoP crossed anterior to the midtarsal joint (when the moment could be calculated)?

lines 61-63 – I still disagree with the authors’ use of ‘quasi-stiffness’ to describe the ankle measurements used by Farley and Morgenroth. Their estimate was based on a single moment in stance phase of a hop, and was not calculated by fitting a slope to the joint angle-moment relationship. Therefore, their measurement does not fit the definition of quasi-stiffness provided at the beginning of this paragraph, and also does not match the calculation used by the authors in this study. Thus, I think the authors should remove ‘quasi’ here to avoid giving readers the false impression that Farley and Morgenroth used a similar method of measuring stiffness as that used in this study.

line 277 – Change ‘Body’ to ‘Added’.

---

## Round 0.4 · accepted · Accept

Thank you very much for carefully addressing all the reviewer comments in your revised version.